# Cognitive Reserve and Its Associations with Pain, Anxiety, and Depression in Patients with Chronic Migraine: A Retrospective Study

**DOI:** 10.3390/jcm14155193

**Published:** 2025-07-22

**Authors:** Yu-Ming Chen, Jen-Hung Wang

**Affiliations:** 1Department of Neurology, Hualien Tzu Chi Hospital, Buddhist Tzu Chi Medical Foundation, Hualien 970, Taiwan; 2School of Medicine, Tzu Chi University, Hualien 970, Taiwan; 3Department of Medical Research, Hualien Tzu Chi Hospital, Buddhist Tzu Chi Medical Foundation, Hualien 970, Taiwan; jenhungwang2011@gmail.com

**Keywords:** cognitive impairment, neurocognitive assessment, headache intensity, psychological distress, psychiatric comorbidity, mental health, quality of life

## Abstract

**Background/Objectives**: Cognitive dysfunction is frequently observed in chronic migraine (CM) patients, but the contributing medical and psychological factors remain unclear. This study investigated associations between the cognitive reserve and medical, psychological, and lifestyle factors in individuals with CM. **Methods**: A retrospective review was conducted at a tertiary referral center in Taiwan. Cognitive function was evaluated via the mini-mental state examination (MMSE), while anxiety and depression were evaluated via the Beck Anxiety and Depression Inventories. Clinical variables included monthly headache days, headache intensity (numerical rating scale), migraine-related disability, and use of preventive medications. Multivariable linear regression analyses were performed to identify independent predictors of the cognitive reserve after adjusting for relevant covariates. **Results:** Among 50 participants (86.0% women; mean age 42.48 ± 13.47 years), six (12.0%) exhibited objective cognitive impairment (MMSE < cutoff). After a covariate adjustment, higher headache intensity was significantly associated with a lower cognitive reserve in anxiety and depression models. Patients with objective cognitive impairment reported significantly higher levels of pain, anxiety, and depression. **Conclusions**: The headache intensity, anxiety, and depression were significantly linked to a lower cognitive reserve in CM patients. These findings highlight the importance of incorporating routine psychological and cognitive assessments in CM care and suggest potential targets for integrative treatment strategies.

## 1. Introduction

Chronic migraine (CM) is a debilitating neurological disorder and among the most disabling forms of migraine. With an estimated global prevalence of 1% to 2%, this condition disproportionately impacts women [1]. According to the International Classification of Headache Disorders, 3rd edition (ICHD-3), CM is diagnosed when a person experiences headaches ≥15 days per month for >3 months, with at least ≥8 days each month meeting migraine criteria or responding to migraine-specific treatment [2]. CM severely impairs individuals’ quality of life and places a substantial burden on healthcare systems and society [1]. In Taiwan, CM substantially contributes to the economic burden through increased healthcare utilization and greater reliance on acute medications [3]. Moreover, this disorder is associated with several comorbidities, including cardiovascular diseases, psychiatric disorders, sleep disorders, and chronic pain syndromes [1,4]. These associations underscore that CM is not merely a headache but a complex, multisystem disorder requiring comprehensive and multidisciplinary care.

Growing evidence indicates that individuals with CM experience notable impairments in cognitive function [5,6,7]. Patients with CM exhibit deficits in attention, memory, and processing speed [8]. Although often overlooked in clinical practice, several studies have confirmed the presence of objective cognitive impairments using standardized neuropsychological evaluations [6,9,10]. The mechanisms underlying cognitive impairment in CM remain unclear; however, they are hypothesized to involve pain-related neural network disruption, psychiatric comorbidities, and the cumulative effects of recurrent migraine attacks [8].

The cognitive difficulties observed in patients with CM may not arise solely from the migraine pathology itself but are likely influenced by a complex interplay of multiple factors. Notably, headache intensity and psychological comorbidities—particularly anxiety and depression—have been shown to exacerbate cognitive dysfunction in individuals with chronic pain conditions [11,12,13]. Psychological factors, particularly anxiety and depression, are closely involved in the pathophysiology of migraine and cognitive functioning. Recent meta-analyses indicate that nearly 50% of patients with CM meet the diagnostic criteria for anxiety or depressive disorders [14], with prevalence rates significantly exceeding those in individuals with episodic migraine or the general population [15]. However, studies specifically examining how these medical and psychological factors interact to influence the cognitive reserve in CM populations (particularly among Asian cohorts) remain scarce. Despite the established associations between these factors and cognitive function, their combined influence on the cognitive reserve—the brain’s capacity to withstand or compensate for neurological damage—remains insufficiently understood in the context of CM.

The cognitive reserve refers to the brain’s ability to adapt to age-related changes, pathological conditions, or injuries without causing cognitive impairment [16,17,18]. A higher cognitive reserve is associated with a reduced risk of cognitive decline and delayed onset of dementia, emphasizing its relevance in dementia-prevention strategies [16]. Despite its importance, there is no universally accepted definition of the cognitive reserve. Some researchers conceptualize it through sociobehavioral proxies, whereas others use residual-based methods or functional neuroimaging to quantify it [18]. In this study, we operationalized the cognitive reserve as the difference between the observed and predicted mini-mental state examination (MMSE) scores based on years of education [19].

This retrospective study examined the relationship between cognitive reserve and various medical, psychological, and lifestyle variables in patients with CM. Although previous research has established associations between CM and cognitive dysfunction, the combined effects of headache intensity, emotional distress, and health behaviors on the cognitive reserve remains unclear—particularly within Asian populations. Hypothetically, a higher headache intensity and more severe symptoms of anxiety and depression would be independently associated with a lower cognitive reserve, even after adjusting for relevant demographic and clinical covariates. Furthermore, to provide clinically relevant insight, we compared the medical, psychological, and lifestyle characteristics of patients with and without objective cognitive impairment. This multidimensional approach advances the current understanding of cognitive vulnerability in CM and identify modifiable targets for early intervention.

## 2. Materials and Methods

### 2.1. Study Design and Ethical Considerations

We conducted a retrospective analysis to examine the association between the cognitive reserve and relevant clinical outcomes in patients with CM. This design was selected based on the availability of standardized, routinely collected clinical data at our tertiary headache clinic. This approach investigated these associations within a real-world clinical context without imposing additional demands on patients. The study protocol was guided by prior literature exploring sleep quality and psychological profiles in CM populations [20], although the exact combination of variables analyzed has not been previously validated as a unified protocol. The study sample was derived from a previously established clinical dataset collected at a tertiary medical center. While this cohort was previously described in an earlier publication addressing a different research objective [21], the current analysis focuses specifically on the cognitive reserve and its associations within the CM population, utilizing distinct outcome variables and statistical approaches. The dataset underwent rigorous methodological processing, including standardized data extraction, independent verification by two researchers, and the resolution of discrepancies through consensus discussions. For this study, particular emphasis was placed on variables relevant to cognitive function and migraine-related clinical features. Ethical approval was obtained from the Research Ethics Committee of Hualien Tzu Chi Hospital, Buddhist Tzu Chi Medical Foundation (IRB113-201-B). The study adhered to the ethical principles outlined in the Declaration of Helsinki.

### 2.2. Study Setting and Population

The present study was carried out at the Headache Clinic of Hualien Tzu Chi Hospital, a tertiary referral center in Taiwan, using retrospective analysis of electronic health records documented between 1 July 2023 and 31 August 2024. The inclusion criteria required participants to be aged 18 years or older and to have received a diagnosis of CM based on the International Classification of Headache Disorders, 3rd edition (ICHD-3). Diagnoses were established by neurologists certified in headache subspecialty care.

Participants were excluded if they had incomplete clinical data, a history of recent trauma or surgery (within the preceding three months), were pregnant or breastfeeding, or had significant neurological or psychiatric comorbidities. Neurological conditions leading to exclusion included cerebrovascular events (e.g., stroke or transient ischemic attack), neurodegenerative diseases such as Parkinson’s disease, multiple sclerosis, or Alzheimer’s disease, epilepsy, intracranial tumors, and traumatic brain injury within the last year. Psychiatric exclusions were based on DSM-5 criteria and encompassed disorders such as schizophrenia spectrum illnesses, bipolar disorder, severe major depressive episodes requiring hospitalization, active substance use disorders, and eating disorders.

Notably, individuals experiencing anxiety or depressive symptoms as a consequence of CM were retained in the study sample. To differentiate migraine-related affective symptoms from primary psychiatric disorders, a temporal assessment was conducted using clinical documentation. A board-certified psychiatrist performed semi-structured evaluations to determine whether the onset of emotional symptoms preceded or followed the development of CM. The severity and nature of psychological complaints were further interpreted in the context of individual headache patterns. Patients who met the diagnostic criteria for psychiatric disorders independent of their migraine history were excluded.

A total of nine individuals were excluded from the original sample of 65 due to missing or incomplete data. Of the remaining 56 eligible participants who met all predefined inclusion criteria, only the first 50 patients—based on the order of data extraction—were included in the final analysis, in accordance with the Institutional Review Board-approved sample size.

### 2.3. Clinical Assessment and Data Collection

The key demographic variables recorded for each participant included age, sex, physical activity patterns, average weekly exercise time, and current use of migraine prophylactic agents. Clinical features were characterized based on the number of monthly headache days (MHDs), cumulative headache days within the past 3 months, and reported pain severity, which was quantified using a numerical rating scale (NRS) [22]. Psychological distress, sleep disturbances, and functional impairments were assessed through standardized self-report instruments administered by licensed clinical psychologists.

Prophylactic treatments were classified into four pharmacological categories: tricyclic antidepressants (e.g., amitriptyline), anticonvulsants (e.g., topiramate), beta-adrenergic blockers (e.g., propranolol), and calcium channel antagonists (e.g., flunarizine). For patients receiving such therapies, details regarding the prescribed agent, dosage, and treatment duration were systematically recorded.

### 2.4. Assessment Tools and Measurements

Migraine-related disability was assessed using the Migraine Disability Assessment (MIDAS) scale, a well-established instrument comprising five items that evaluate the degree of functional impairment due to migraine over a three-month period [23]. Total scores were used to classify the level of disability into four categories: 0–5 (none or minimal), 6–10 (mild), 11–20 (moderate), and ≥21 (severe). To further quantify the migraine burden, the cumulative number of headache days during the preceding three months was also documented.

Psychological distress was assessed using two standardized self-report instruments: anxiety was measured using the Beck Anxiety Inventory (BAI), including 21 items scored on a 0–3 scale and yielding a total score of 0–63. Severity levels were interpreted as follows: 0–7 (minimal), 8–15 (mild), 16–25 (moderate), and ≥26 (severe) [24]. A threshold score of ≥16 was used to identify clinically relevant anxiety symptoms. Depressive symptoms were evaluated using the 21-item Beck Depression Inventory (BDI), with total scores of 0–13 (minimal), 14–19 (mild), 20–28 (moderate), and ≥29 (severe) [25]. A score of ≥20 implied clinically relevant depression.

Sleep quality was measured using the Chinese version of the Pittsburgh Sleep Quality Index (PSQI) [26]—a widely used tool that assessing multiple dimensions of sleep over the past month. The PSQI comprises seven domains: perceived sleep quality, time to fall asleep, total sleep duration, habitual sleep efficiency, sleep-related disturbances, the use of sleep medications, and daytime dysfunction. Each domain is rated on a 0–3 scale, yielding a total score between 0 and 21, with higher scores reflecting poorer sleep quality. Herein, a global PSQI score of ≥6 was used to indicate poor sleep quality [26].

Cognitive status was assessed using the Chinese version of the MMSE [19]—a standardized instrument frequently employed for cognitive screening in clinical settings. The MMSE evaluates five key domains: temporal and spatial orientation, immediate memory (registration), attention and calculation, delayed recall, and language function. Scores range from 0 to 30, with higher values reflecting greater cognitive ability [19].

The primary outcome of this study was cognitive reserve, which was operationalized as a proxy measure derived from the difference between the observed and predicted MMSE scores based on years of education. Predicted MMSE values were estimated using normative reference data derived from [19], and the resulting difference score (MMSE_observed − MMSE_predicted) was treated as a continuous variable in subsequent analyses. Higher scores reflected better-than-expected cognitive performance relative to demographic norms.

### 2.5. Sample Size Calculation and Power Analysis

Sample size estimation was performed using G*Power software (version 3.1.9.2, Heinrich-Heine University, Düsseldorf, Germany) [27]. A priori power analysis was performed to calculate the minimum number of participants required for conducting multivariable linear regression, identifying factors associated with cognitive reserve in individuals diagnosed with CM. The analysis was based on the following assumptions: a medium effect size (f^2^ = 0.32), an alpha level of 0.05, statistical power of 0.80, and six predictor variables. Under these parameters, the estimated minimum sample size was 50 participants. This sample size also aligns with commonly recommended methodological standards, which suggest recruiting 5 to 10 participants per predictor variable [28], thereby supporting the stability and interpretability of the regression model.

### 2.6. Statistical Analysis

All statistical procedures were performed using Statistical Package for the Social Sciences (version 25.0; IBM Corp., Armonk, NY, USA). Statistical significance was set at a two-tailed *p*-value threshold of <0.05. The distributional normality of continuous variables was evaluated using the Kolmogorov–Smirnov test, with a *p*-value > 0.05 indicating an approximately normal distribution [29].

Variables were categorized as continuous or dichotomous. Continuous variables included age, weekly exercise duration, PSQI scores, headache intensity measured using the NRS, MHDs, total headache days over the past 3 months, MIDAS scores, BAI scores, BDI scores, MMSE scores, and cognitive reserve scores. Categorical variables included sex (man or woman), regular exercise (yes or no), and the use of preventive medications (yes or no). While standard clinical cutoff scores are available for the BAI and BDI, these scales were treated as continuous variables in correlation and regression analyses to enhance statistical sensitivity and preserve the full range of symptom variation.

Continuous data are summarized as means and standard deviations, while categorical data are reported as counts and corresponding percentages. The statistical analysis followed a three-phase approach. First, group differences in key clinical and psychological variables were examined between participants with and without objective cognitive impairment using the independent sample *t*-test. Second, pairwise relationships between the cognitive reserve and each predictor variable were examined using Pearson’s correlation analysis, with the correlation strength interpreted as strong (*r* > 0.60), moderate (*r* = 0.30–0.60), or weak (*r* < 0.30), following established criteria [30]. Third, multivariate linear regression models were subsequently applied to identify variables independently associated with the cognitive reserve. The models included demographic (age and sex), clinical (sleep quality, headache intensity, and MIDAS score), and psychological (anxiety and depression symptomatology) variables. Variance inflation factors (VIFs) were used to assess multicollinearity, with values <10 considered acceptable. Standardized beta coefficients, 95% confidence intervals (95% CIs), *p*-values, and adjusted *R*^2^ values were used to interpret the effect size and model performance.

Considering the strong correlation between anxiety and depression scores (*r* > 0.60), two separate multivariable regression models were developed to mitigate potential multicollinearity. Model 1 included anxiety (BAI) along with clinical covariates: age, sex, PSQI score, headache intensity, and MIDAS score. Model 2 substituted depression (BDI) for anxiety while retaining the same set of covariates. This dual-model strategy aligns with established best practices for addressing collinearity between overlapping affective variables in clinical research settings [31].

### 2.7. Use of GenAI in Writing

We used a generative artificial intelligence (AI) tool (ChatGPT-4o, OpenAI) to assist in improving the fluency and clarity of the manuscript, focusing on grammar, spelling, and sentence structure. We critically reviewed, edited, and approved all AI-assisted content, and take full responsibility for the accuracy, originality, and integrity of the manuscript.

## 3. Results

### 3.1. Baseline Demographic and Clinical Data

After excluding nine patients with incomplete data from the initial sample of 65 patients, the first 50 of the remaining 56 eligible patients were analyzed, according to the IRB-approved sample size.

Fifty patients with CM were enrolled, of whom six (12.0%) were identified as having objective cognitive impairment based on MMSE scores (Table 1). The cognitively impaired and non-impaired groups did not significantly differ in age, sex distribution, exercise habits, sleep disturbance, or preventive medication use.

Participants with objective cognitive impairment reported a significantly higher headache intensity (8.50 ± 1.87 vs. 5.84 ± 1.75, *p* = 0.001), greater anxiety (BAI: 31.50 ± 12.57 vs. 15.95 ± 11.84, *p* = 0.004), and more severe depressive symptoms (BDI: 34.33 ± 17.83 vs. 17.16 ± 11.93, *p* = 0.003) than those without objective cognitive impairment (Table 1 and Figure 1). As expected, MMSE scores were significantly lower in the objective cognitive impairment group (21.17 ± 1.72 vs. 28.43 ± 1.50, *p* < 0.001), confirming the presence of objective cognitive deficits in this subgroup. Among patients with CM, six individuals were identified as having objective cognitive impairment. Predominantly affected cognitive domains included sustained attention, short-term memory, calculation, temporal orientation, and visuospatial construction.

### 3.2. Associations Among Clinical Parameters

As shown in Table 2, the Pearson correlation analysis identified multiple significant relationships among clinical and psychological measures. Cognitive performance, as measured based on MMSE scores, was negatively correlated with headache intensity (NRS: *r* = −0.424, *p* < 0.01), anxiety symptoms (BAI: *r* = −0.336, *p* < 0.05), and depressive symptoms (BDI: *r* = −0.343, *p* < 0.05). These findings indicate statistically significant associations between symptom severity and lower MMSE performance.

Headache intensity (NRS) was positively correlated with MHDs (*r* = 0.374, *p* < 0.01) and migraine-related disability (MIDAS: *r* = 0.382, *p* < 0.01). The MIDAS was also moderately correlated with MHDs (*r* = 0.306, *p* < 0.05).

Sleep quality, as measured based on the PSQI, showed significant positive correlations with anxiety symptoms (BAI: *r* = 0.532, *p* < 0.01) and depressive symptoms (BDI: *r* = 0.492, *p* < 0.01). Age was negatively correlated with anxiety (BAI: r = −0.284, *p* < 0.05) and depression (BDI: *r* = −0.298, *p* < 0.05). The anxiety and depression scores were also highly correlated (*r* = 0.830, *p* < 0.01).

No significant correlations were observed between weekly exercise duration and any clinical variable, including headache intensity, cognitive function (MMSE), mood (BAI and BDI), or sleep quality (PSQI) (all *p* > 0.05).

### 3.3. Associations Between Clinical Variables and Cognitive Reserve

Multivariable linear regression analyses were performed to identify predictors of the cognitive reserve (Table 3). In unadjusted analyses, greater headache intensity (NRS: β = −0.60, 95% CI: −0.94 to −0.26, *p* = 0.001), anxiety symptoms (BAI: β = −0.08, 95% CI: −0.14 to −0.03, *p* = 0.003), and depressive symptoms (BDI: β = −0.08, 95% CI: −0.13 to −0.03, *p* = 0.002) were significantly associated with a lower cognitive reserve.

In adjusted Model 1 (R^2^ = 0.38), which accounted for age, sex, and preventive medication usage, three variables remained significant: age (β = −0.06, 95% CI: −0.78 to −0.01, *p* = 0.026), NRS (β = −0.47, 95% CI: −0.84 to −0.10, *p* = 0.015), BAI (β = −0.08, 95% CI: −0.13 to −0.02, *p* = 0.005). In Model 2 (R^2^ = 0.37), which substituted BAI with BDI to address collinearity, BDI remained a significant predictor (β = −0.07, 95% CI: −0.12 to −0.02, *p* = 0.008). In both models, sex and preventive medication usage were not significant predictors (all *p* > 0.05). VIFs for all variables were <10, indicating no multicollinearity concerns.

Figure 2 presents Scatterplots illustrating the associations between the cognitive reserve and key predictors. Headache intensity (NRS), anxiety (BAI), and depressive symptoms (BDI) all exhibited negative linear trends with cognitive reserve, consistent with the regression results. Each plot illustrates the direction and strength of the association between the respective predictor and cognitive reserve scores.

## 4. Discussion

The present findings provide valuable evidence regarding the complex interplay between the cognitive reserve and CM. Specifically, higher headache intensity, anxiety, and depressive symptoms emerged as key factors associated with a lower cognitive reserve. These outcomes expand the previous studies by demonstrating the presence of cognitive deficits in patients with CM and the psychological and pain-related predictors of lower cognitive resilience. Compared with earlier literature (Table 4), largely focused on general cognitive impairment or used comprehensive neuropsychological batteries, our study uniquely operationalized the cognitive reserve and quantified its association with clinically relevant variables. These findings underscore the need for a multidisciplinary clinical approach addressing psychological burden and headache intensity while managing CM.

The notable association between anxiety and depression scores in this cohort reflects the substantial overlap often observed between these psychological conditions [34,35]. Our regression analyses identified headache intensity, anxiety, and depression as significant predictors of a lower cognitive reserve, each remaining statistically significant in independent models following an adjustment for demographic and clinical covariates. These findings suggest that multiple psychosocial factors independently contribute to cognitive vulnerability. This finding supports and extends previous evidence that anxiety and depression can affect cognitive function [36]. In line with our regression findings, individuals with objective cognitive impairment exhibited markedly higher levels of headache intensity, anxiety, and depressive symptoms (Figure 1). These between-group differences suggest that an elevated symptom burden is closely linked to lower cognitive reserve in patients with CM. The observed patterns are consistent with those of previous reports indicating that emotional distress and headache intensity are key contributors to cognitive dysfunction in chronic pain populations [11,12,13].

Herein, patients with CM and objective cognitive impairment exhibited deficits in attention, short-term memory, calculation, orientation, and visuoconstruction tasks based on the MMSE. These findings reflect various cognitive challenges, particularly in attentional control and short-term memory. Previous research suggests that individuals with CM often exhibit impaired cognitive function [5,6,7], particularly in areas such as cognitive flexibility [37,38,39], task switching, and executive functioning [40]. CM is more strongly associated with cognitive complaints and executive dysfunction than episodic migraine [39,40]. These impairments may be partly due to disruptions in the functional organization of brain networks responsible for pain regulation and cognitive control, which could contribute to a lower cognitive reserve in this population. De Tommaso et al. [41] found that patients with episodic and CM demonstrated the diminished suppression of laser-evoked potential amplitudes during cognitive tasks performed under acute pain, suggesting the impaired cognitive modulation of nociceptive signals. Furthermore, functional magnetic resonance imaging studies have identified attenuated activation and deactivation patterns in brain regions typically engaged in cognitive tasks, such as the left dorsolateral prefrontal cortex, dorsal anterior midcingulate cortex, and cerebellum [42]. Unlike healthy controls—who show marked task-related deactivation in these regions that further diminishes under pain—patients with migraine exhibit reduced baseline deactivation and limited modulation in response to acute pain stimuli.

These findings support the hypothesis that CM impairs the dynamic interaction between pain- and cognition-related neural systems, particularly those governing executive functions and attentional control. Such disruptions may help explain why headache intensity emerged as an independent predictor of lower cognitive reserve in our study, beyond the contributions of anxiety and depression.

In addition, CM over time has been linked to structural disruptions in right-lateralized anterior white matter tracts, which are functionally related to cognitive control, effect regulation, pain perception, and resilience [43]. These structural changes may contribute to cognitive decline or reduced cognitive reserve in this population [32]. The involvement of shared neural pathways may help explain the heightened cognitive vulnerability observed in individuals with CM, especially in the context of persistent pain or emotional distress [32].

The demographic characteristics of our sample align with well-documented epidemiological trends in CM, suggesting that the cohort is representative of the broader clinical population. This investigation focused on exploring how headache intensity and psychological comorbidities relate to cognitive performance within this patient group. The mean MIDAS score of 56.76 ± 57.74 reflects a considerable level of migraine-related disability. Preventive medication use was reported by just 24.0% of the participants, substantially lower than international norms [44,45], indicating a clear gap in migraine management and the need for improved preventive care. The impact of preventive medications on cognition merits special consideration in studies on cognitive function among patients with migraine. Certain migraine prophylactics, such as topiramate, amitriptyline, and flunarizine, have well-documented cognitive side effects, including drowsiness, impaired attention, impaired decision-making, and reduced processing speed [46]. The cognitive effects of preventive medications represent a potential confounder that may explain some inconsistent cognitive results in the literature [8].

To address this potential confounding factor, preventive medications were categorized based on their known cognitive side-effect profiles. In particular, they were classified into two groups: (1) medications with documented cognitive effects—such as topiramate, amitriptyline, and flunarizine; and (2) those with minimal or no established cognitive impact—such as propranolol and valproate. Considering the relatively small proportion of participants using preventive medications (24.0%), a dichotomous variable was incorporated into the regression model to indicate whether a participant was taking a medication with potential cognitive side effects. This strategy enabled statistical control for medication-related influences on cognitive reserve outcomes.

This study provides important insights into the clinical and theoretical implications of cognitive reserve in patients with CM. First, our findings suggest that headache intensity, anxiety, and depression are significant predictors of a lower cognitive reserve in this population. These findings underscore the importance of routinely assessing psychological symptoms in patients with CM, as they may play a critical role in lower cognitive reserve. Our work bridges the gap between general chronic pain literature [11,12,13] and migraine-specific research by exploring the associations between CM and cognitive reserve.

We also advocate for the routine implementation of standardized psychological screening instruments, such as the BAI and BDI, in the clinical assessment of patients with CM. The proactive use of these tools can support the early detection of individuals at an elevated risk of cognitive impairment, thereby allowing for timely and targeted interventions.

Finally, our analysis revealed the proportion of patients experiencing objective cognitive impairment, highlighting the critical role of cognitive assessments in this population. In particular, the MMSE is an efficient tool for routine cognitive screening because of its simplicity, cost-effectiveness, and ability to detect cognitive changes over time. In addition to routine use, the MMSE can facilitate the early identification of cognitive problems. For instance, when a patient’s observed MMSE score deviates significantly from the predicted threshold—indicating lower cognitive reserve—it may serve as an early warning sign of potential cognitive decline. This approach enables proactive monitoring and timely intervention, particularly in patients with either subjective or objective cognitive complaints.

This study offers several notable strengths. First, by examining an Asian cohort, it addresses a critical gap in the existing literature [47,48] because most prior research on migraine and cognitive function has primarily focused on Western populations. Accordingly, this study offers culturally specific perspectives on how headache intensity and psychological comorbidities interact to influence cognitive outcomes. Second, we used validated assessment instruments, including the MMSE with education-adjusted cutoff values, which enabled the more accurate identification of objective cognitive impairment while accounting for demographic variability. This methodological rigor enhances the reliability and clinical relevance of our results. Third, we controlled for the potential confounding effects of preventive medication use by incorporating this variable into our regression analyses—an aspect frequently overlooked in similar studies. By adjusting for these factors, our analysis provides a more nuanced understanding of the complex interactions among pain, emotional distress, and the cognitive reserve in CM patients.

Despite these strengths, this study has several limitations that should be acknowledged. The single-center setting and relatively modest sample size may limit the generalizability of our findings to the broader CM population, particularly considering that our tertiary care context may involve a higher proportion of complex or treatment-resistant cases. Due to its retrospective nature, the study was not designed to capture dynamic clinical changes, such as fluctuations in medication use or symptom progression over time. While the sample size was appropriate for the primary analyses conducted, the interpretation of subgroup findings and models with multiple covariates should be made with caution. Although we attempted to adjust for the cognitive effects of preventive medications, the limited number of patients receiving such treatments restricted our ability to conduct detailed analyses according to the drug type, dosage, and treatment duration. Moreover, the retrospective cross-sectional nature of the study prevents any causal inferences regarding the observed associations among headache intensity, psychological symptoms, and the cognitive reserve. A longitudinal, multicenter study with expanded and more heterogeneous samples is warranted to validate and extend these findings. To better elucidate the complex interactions among migraine, cognitive function, and environmental context, such studies should incorporate comprehensive pharmacological data and detailed sociodemographic information, including marital status, employment status, household income, and living conditions. Furthermore, the questionnaires used for cognitive and psychological assessment were extracted from routine clinical records. Although BAI and BDI were administered under our institution’s active license, the MMSE was based on a historically used Chinese version frequently adopted in Taiwan before the official licensing of the authorized edition in July 2025. No test items were reproduced, and all data were anonymized and analyzed according to ethical and institutional standards. Finally, because the study did not include post-treatment follow-up, we could not determine whether cognitive impairment improved alongside clinical outcomes. Future longitudinal studies are required to explore the reversibility of cognitive dysfunction in response to preventive medication.

## 5. Conclusions

The findings of this study demonstrate that headache intensity and psychological symptoms—particularly anxiety and depression—are significantly associated with a lower cognitive reserve in individuals with CM. Patients with objective cognitive impairment (MMSE_observed < cutoff) reported markedly higher levels of pain, anxiety, and depressive symptoms than those without cognitive deficits. These results highlight the clinical importance of incorporating routine psychological and cognitive evaluations into CM management. The observed association between coexisting pain and psychological distress with a diminished cognitive reserve further emphasizes the need for an integrative, multidisciplinary approach.

## Figures and Tables

**Figure 1 jcm-14-05193-f001:**
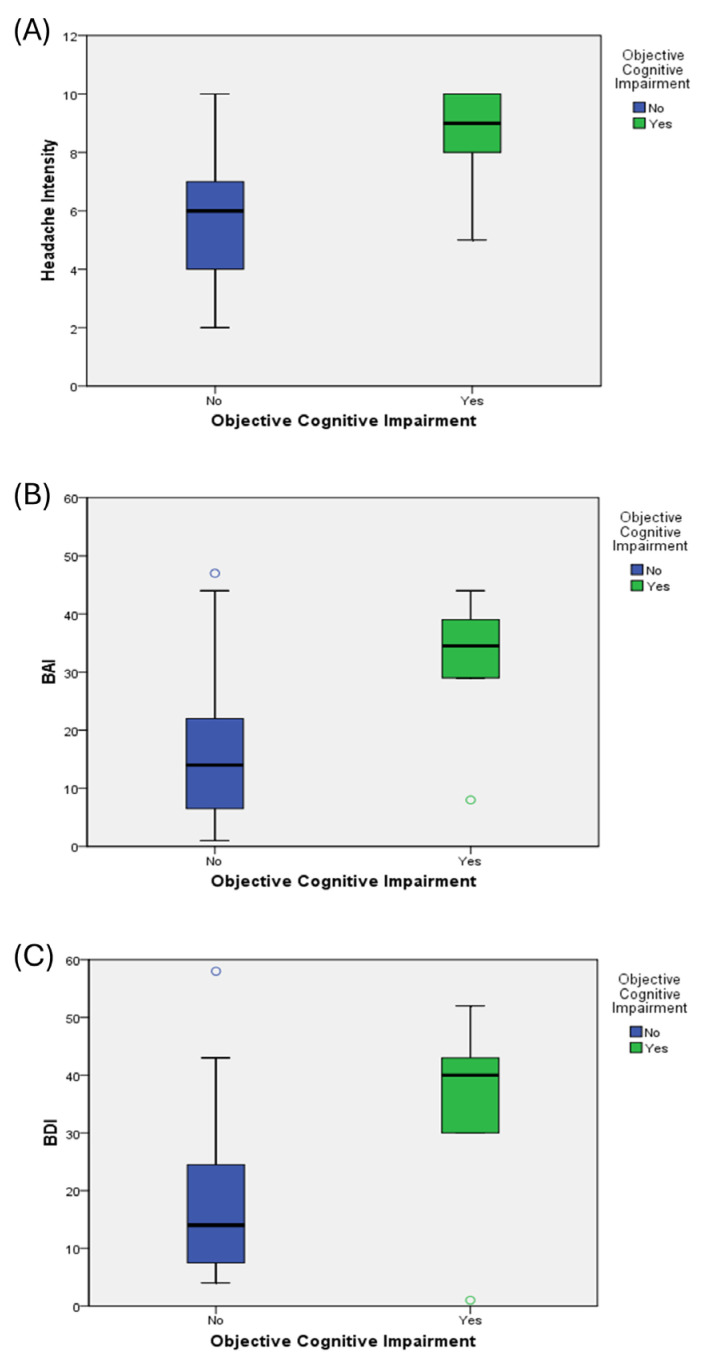
Group differences in headache intensity and psychological symptom severity between patients with chronic migraine with and without objective cognitive impairment. Box plots present (**A**) headache intensity (measured by the NRS), (**B**) the BAI scores, and (**C**) the BDI scores according to cognitive status (yes vs. no). The central line within each box indicates the median value; box boundaries represent the IQR, and whiskers extend to the furthest data points within 1.5 × IQR. Data points exceeding this range are plotted individually as open circles to denote outliers. Outliers are depicted using unfilled circles. Abbreviations: NRS, numerical rating scale; BAI, Beck Anxiety Inventory; BDI, Beck Depression Inventory; IQR, interquartile range.

**Figure 2 jcm-14-05193-f002:**
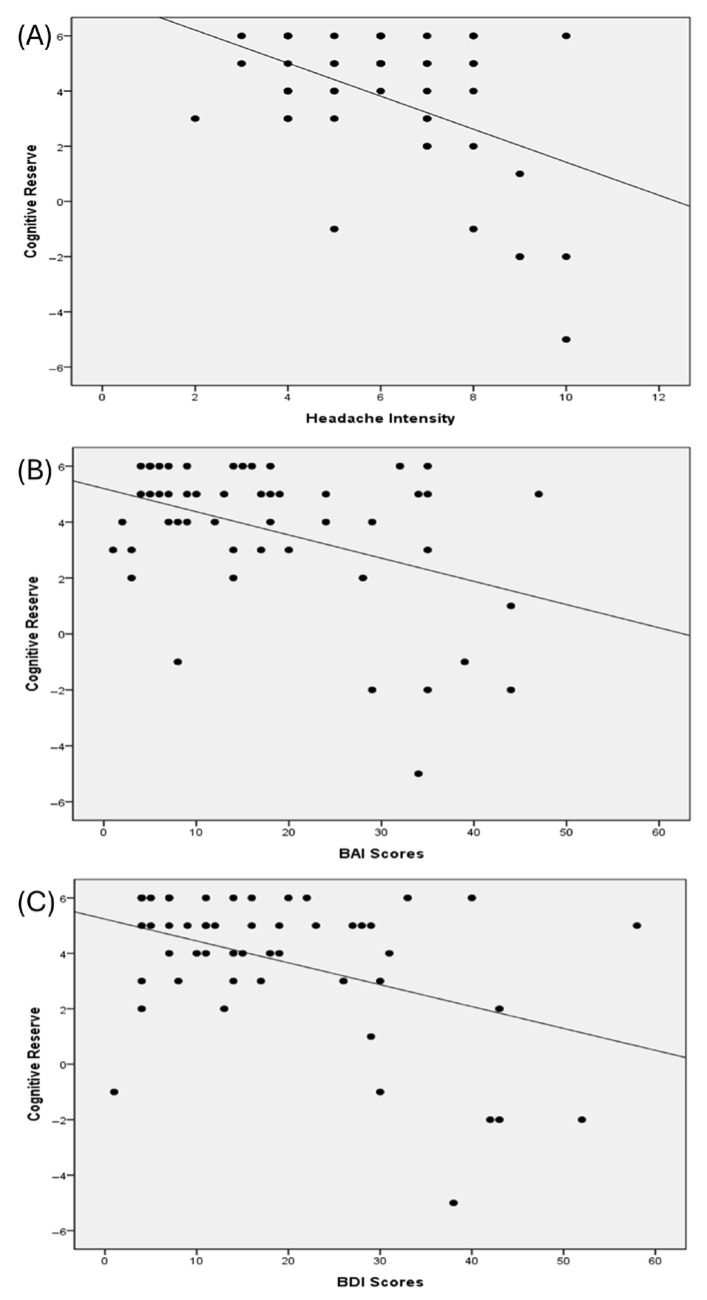
Scatter plots illustrating the associations between cognitive reserve and (**A**) headache intensity (measured by the NRS), (**B**) anxiety severity (the BAI scores), and (**C**) depressive symptoms (the BDI scores) in patients with chronic migraine. Linear regression lines are included to indicate the direction and strength of each relationship. Cognitive reserve was defined as residual scores derived from MMSE values adjusted for years of education. Abbreviations: NRS, numerical rating scale; BAI, Beck Anxiety Inventory; BDI, Beck Depression Inventory; MMSE, mini-mental status examination.

**Table 1 jcm-14-05193-t001:** Characteristics of patients with chronic migraine according to cognitive impairment status (N = 50).

Item	Objective Cognitive Impairment			*p*-Value
	No	Yes	Total	
N	44	6	50	
Age (years)	42.50 ± 13.50	42.33 ± 14.54	42.48 ± 13.47	0.978
Sex				0.576
Women	37 (84.1%)	6 (100.0%)	43 (86.0%)	
Men	7 (15.9%)	0 (0.0%)	7 (14.0%)	
Regular exercise (%)	16 (36.4%)	1 (16.7%)	17 (34.0%)	0.650
Weekly exercise duration (mins)	96.86 ± 234.70	23.33 ± 57.15	88.04 ± 221.93	0.452
PSQI score	11.41 ± 5.26	15.17 ± 6.55	11.86 ± 5.50	0.117
Insomnia(PSQI ≥ 6) (%)	36 (81.8%)	5 (83.3%)	41 (82.0%)	1.000
Preventive medication use (%)	11 (25.0%)	1 (16.7%)	12 (24.0%)	1.000
Preventive medication category				1.000
Used preventive medications	11 (25.0%)	1 (16.7%)	12 (24.0%)	
Used medications potentially affecting cognition	6 (13.6%)	0 (0.0%)	6 (12.0%)	
Used medications with less cognitive impact	5 (11.4%)	1 (16.7%)	6 (12.0%)	
Did not use preventive medications	33 (75.0%)	5 (83.3%)	38 (76.0%)	
Headache intensity (NRS)	5.84 ± 1.75	8.50 ± 1.87	6.16 ± 1.95	0.001 *
Monthly headache days	22.70 ± 6.77	25.83 ± 6.65	23.08 ± 6.77	0.293
Headache days over past 3 months	70.95 ± 23.72	89.17 ± 2.04	73.14 ± 23.02	0.069
MIDAS score	52.89 ± 54.54	85.17 ± 77.46	56.76 ± 57.74	0.202
BAI score	15.95 ± 11.84	31.50 ± 12.57	17.82 ± 12.85	0.004 *
BDI score	17.16 ± 11.93	34.33 ± 17.83	19.22 ± 13.75	0.003 *
MMSE score (observed)	28.43 ± 1.50	21.17 ± 1.72	27.56 ± 2.82	<0.001 *

Abbreviations: PSQI, Pittsburgh Sleep Quality Index; NRS, numerical rating scale; MHD, monthly headache days; MIDAS, Migraine Disability Assessment Scale; BAI, Beck Anxiety Inventory; BDI, Beck Depression Inventory; MMSE, mini-mental state examination. Data are presented as means ± standard deviations for continuous variables (age, PSQI, weekly exercise duration, NRS, MHD, MIDAS, BAI, BDI, and MMSE) and n (percentage) for categorical variables (sex, preventive medication usage, and exercise habit). *p*-values represent comparisons between participants with and without objective cognitive impairment using independent *t*-tests for continuous variables and the chi-square test or Fisher’s exact test for categorical variables. * indicates statistical significance (*p* < 0.05).

**Table 2 jcm-14-05193-t002:** Pearson correlations among clinical and cognitive variables (N = 50).

Correlation	Age	MHD	NRS	MIDAS	Weekly Exercise Duration (min)	BAI	BDI	MMSE (Observed)	PSQI
Age	1	0.051	−0.012	−0.060	0.039	−0.284 *	−0.298 *	−0.242	−0.147
MHD		1	0.374 **	0.306 *	−0.011	0.128	0.107	−0.248	0.180
NRS			1	0.382 **	−0.256	0.299 *	0.373 **	−0.424 **	0.282 *
MIDAS				1	−0.094	0.248	0.192	−0.207	0.243
Weekly exercise duration (min)					1	−0.039	−0.011	0.035	0.043
BAI						1	0.830 **	−0.336 *	0.532 **
BDI							1	−0.343 *	0.492 **
MMSE (observed)								1	−0.174
PSQI									1

Abbreviations: PSQI, Pittsburgh Sleep Quality Index; NRS, numerical rating scale; MHD, monthly headache days; MIDAS, Migraine Disability Assessment Scale; BAI, Beck Anxiety Inventory; BDI, Beck Depression Inventory; MMSE, mini-mental state examination. Data are presented as Pearson correlation coefficients. * *p* < 0.05; ** *p* < 0.01 indicate statistical significance.

**Table 3 jcm-14-05193-t003:** Multivariate model of cognitive reserve determinants in chronic migraine (N = 50).

	Crude		Adjusted (Model 1), R^2^ = 0.38			Adjusted (Model 2), R^2^ = 0.37		
	β (95% CI)	*p*-Value	β (95% CI)	*p*-Value	VIF	β (95% CI)	*p*-Value	VIF
Age	−0.03 (−0.08, 0.02)	0.277	−0.06 (−0.11, −0.01)	0.026 *	1.178	−0.06 (−0.11, −0.01)	0.026 *	1.193
Sex (man vs. woman)	0.82 (−1.29, 2.94)	0.438	−0.27 (−2.16, 1.62)	0.772	1.132	−0.26 (−2.16, 1.64)	0.785	1.132
Preventive medication usage (yes vs. no)	0.26 (−1.47, 1.99)	0.765	0.84 (−0.64, 2.32)	0.261	1.053	0.76 (−0.73, 2.25)	0.308	1.052
MHD	−0.10 (−0.20, 0.01)	0.075						
NRS	−0.60 (−0.94, −0.26)	0.001 *	−0.47 (−0.84, −0.10)	0.015 *	1.345	−0.41 (−0.79, −0.02)	0.038 *	1.435
MIDAS	−0.01 (−0.02, 0.001)	0.081	−0.001 (−0.013, 0.01)	0.807	1.225	−0.003 (−0.015, 0.009)	0.585	1.201
BAI	−0.08 (−0.14, −0.03)	0.003 *	−0.08 (−0.13, −0.02)	0.005 *	1.230			
BDI	−0.08 (−0.13, −0.03)	0.002 *				−0.07 (−0.12, −0.02)	0.008 *	1.294

Abbreviations: MHD, monthly headache days; NRS, numerical rating Scale; MIDAS, Migraine Disability Assessment Scale; BAI, Beck Anxiety Inventory; BDI, Beck Depression Inventory; MMSE, mini-mental state examination; VIF, variance inflation factor. Data are presented as standardized regression coefficients (β) with 95% confidence intervals. * *p* < 0.05 was considered statistically significant.

**Table 4 jcm-14-05193-t004:** Comparison of current findings with prior studies on chronic migraine. Arrow symbols (↑) indicate higher levels or scores; (↓) indicates lower levels or scores; (–) indicates no significant association. Previous studies consistently report cognitive impairment in patients with CM, but vary in whether anxiety, depression, or headache intensity independently predict cognitive outcomes. Our study uniquely focused on the cognitive reserve and showed that headache intensity and psychological distress are independently associated with a lower cognitive reserve, highlighting the value of a multidimensional assessment approach.

Study (Year)	Sample (CM)	Cognitive Findings	Headache Intensity/Disability	Psychological Factors (Anxiety/Depression)
Present study, 2025	N = 50 CM (86% women); MMSE used for cognition	12% met criteria for objective cognitive impairment (MMSE < cutoff) **↑** BAI/BDI scores in cognitively impaired group	**↑** headache intensity correlated with **↓** CR	**↑** anxiety and depression associated with **↓** CR
Gómez-Beldarrain et al., 2015 [32]	N = 18 CM–MOH; 90.9% women; CR index used for cognition	**↓** CR in CM–MOH group; **↓** CR associated with **↓** QoL	– association with headache intensity not analyzed	↑ anxiety and depression in CM–MOH; ↑ CR associated with ↓ anxiety and depression
Ferreira et al., 2018 [6]	N = 30 CM vs. 30 controls; comprehensive neuropsychological battery (MoCA, Stroop, etc.)	**↓** cognition in CM; CM = independent risk factor	– association with headache intensity not analyzed	– cognitive deficits occurred regardless of depression/anxiety or preventive medication use
Latysheva et al., 2020 [7]	N = 144 CM vs. 44 EM; neuropsychological tests (MoCA, RAVLT, etc.) and HADS for mood	**↑** cognitive deficits in CM vs. EM; predictors: CM diagnosis, low education	– CM (i.e., ↑ frequency) linked to ↓ cognitive performance; headache intensity not analyzed	– no independent effect of anxiety/depression on cognition after adjustment
Lozano-Soto et al., 2023 [33]	N = 39 CM vs. 20 controls; full neuropsychological tests during interictal period	54% had neuropsychological impairment;	↑ headache intensity/disability severity predicted ↓ global cognitive performance	– no independent effect of anxiety/depression on cognition after adjustment

Abbreviations: BAI, Beck Anxiety Inventory; BDI, Beck Depression Inventory; CM, chronic migraine; CM–MOH = chronic migraine with medication overuse headache; CR, cognitive reserve; EM, episodic migraine; HADS, Hospital Anxiety and Depression Scale; MMSE, mini-mental state examination; MoCA, Montreal Cognitive Assessment; QoL, quality of life; RAVLT, Rey Auditory Verbal Learning Test.

## Data Availability

Data are available from the corresponding author upon reasonable request.

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
