# Peer review of "Cognitive Reserve and Its Associations with Pain, Anxiety, and Depression in Patients with Chronic Migraine: A Retrospective Study"

_jcm, 2025, doi:10.3390/jcm14155193_

Round 1
Reviewer 1 Report
Comments and Suggestions for Authors
The authors described the relationship between migraine, especially chronic migraine and cognitive disfunction. It’s an interesting and hot topic.
However, this manuscript has some questionable issues.
First of all, in chronic migraine patients (CM), what parts of the cognitive function did they perform poorly on the MMSE? What kind of cognitive impairment did they have? I understand that it is due to pain, anxiety, and psychological factors.
Second, If CM improves, will the MMSE and cognitive impairment improve? Does CM make people more likely to develop dementia such as Alzheimer's disease in the future?
Author Response
"Please see the attachment."

Reviewer 2 Report
Comments and Suggestions for Authors
Firstly, would like to thank the authors for submitting their valuable work to our Journal. Below, a few suggestions for your article”
- Please observe author guidelines for JCM and review previous studies published in the journal, restructure the text, title, and abstract accordingly
- Would recommend reviewing study design and methodology with an epidemiologist and statistitian. Given that there have been several previous studies published on the same subject, would advise against exploratory approach
- In the introduction, I would recommend citing the official diagnostic criteria for migraines that are mentioned on the international classification for headaches website (ICHD-3).
- Would further specify in the introduction as well as rest of the text specific pain syndrome the authors intended to investigate as saying “moderate to severe pain” is very vague
- Would further clarify the specific goals of the study and study hypothesis since there have been several similar studies in the literature recently. Authors must provide details as to what is new and original that they are proposing to study.
- Please follow STROBE guidelines and attach a point by point description as to how each item in the STROBE guideline was addressed in your study
- Please re-structure the study according to the STROBE guidelines. including re-structuring intro, abstract, methods, results, and discussion
- Explain in the methods how the authors chose this particular study design and protocol, if it was previously validated, inclusion and exclusion criteria
- Since there are several studies published in the literature on the same subject would recommend adding a comparison table with findings from your study and comparing to previous literature
- Please further clarify why each particular test was chosen in your study, if it was previously validated for your particular population and how, and if copyrights were respected
Mild changes required
Author Response
"Please see the attachment."

Round 2
Reviewer 1 Report
Comments and Suggestions for Authors
This revised manuscript is suitable for this journal.